# Transcriptional Reprogramming of *Candida tropicalis* in Response to Isoespintanol Treatment

**DOI:** 10.3390/jof9121199

**Published:** 2023-12-15

**Authors:** Orfa Inés Contreras-Martínez, Alberto Angulo-Ortíz, Gilmar Santafé-Patiño, Katia Aviña-Padilla, María Camila Velasco-Pareja, María Fernanda Yasnot

**Affiliations:** 1Biology Department, Faculty of Basic Sciences, University of Córdoba, Montería 230002, Colombia; 2Chemistry Department, Faculty of Basic Sciences, University of Córdoba, Montería 230002, Colombia; aaangulo@correo.unicordoba.edu.co (A.A.-O.); gsantafe@correo.unicordoba.edu.co (G.S.-P.); 3Center for Research and Advanced Studies of the I.P.N. Unit Irapuato, Irapuato 36821, Mexico; ib.katia@gmail.com; 4Bacteriology Department, Faculty of Health Sciences, University of Córdoba, Montería 230002, Colombia; mariavelascop@correo.unicordoba.edu.co (M.C.V.-P.); myasnot@correo.unicordoba.edu.co (M.F.Y.)

**Keywords:** transcriptomics, *Candida tropicalis*, isoespintanol, genetic dysregulation, mitochondria, cell wall

## Abstract

*Candida tropicalis*, an opportunistic pathogen, ranks among the primary culprits of invasive candidiasis, a condition notorious for its resistance to conventional antifungal drugs. The urgency to combat these drug-resistant infections has spurred the quest for novel therapeutic compounds, with a particular focus on those of natural origin. In this study, we set out to evaluate the impact of isoespintanol (ISO), a monoterpene derived from *Oxandra xylopioides*, on the transcriptome of *C. tropicalis*. Leveraging transcriptomics, our research aimed to unravel the intricate transcriptional changes induced by ISO within this pathogen. Our differential gene expression analysis unveiled 186 differentially expressed genes (DEGs) in response to ISO, with a striking 85% of these genes experiencing upregulation. These findings shed light on the multifaceted nature of ISO’s influence on *C. tropicalis*, spanning a spectrum of physiological, structural, and metabolic adaptations. The upregulated DEGs predominantly pertained to crucial processes, including ergosterol biosynthesis, protein folding, response to DNA damage, cell wall integrity, mitochondrial activity modulation, and cellular responses to organic compounds. Simultaneously, 27 genes were observed to be repressed, affecting functions such as cytoplasmic translation, DNA damage checkpoints, membrane proteins, and metabolic pathways like trans-methylation, trans-sulfuration, and trans-propylamine. These results underscore the complexity of ISO’s antifungal mechanism, suggesting that it targets multiple vital pathways within *C. tropicalis*. Such complexity potentially reduces the likelihood of the pathogen developing rapid resistance to ISO, making it an attractive candidate for further exploration as a therapeutic agent. In conclusion, our study provides a comprehensive overview of the transcriptional responses of *C. tropicalis* to ISO exposure. The identified molecular targets and pathways offer promising avenues for future research and the development of innovative antifungal therapies to combat infections caused by this pathogenic yeast.

## 1. Introduction

Candidemias represent a significant healthcare challenge, ranking among the foremost causes of morbidity and mortality, particularly in patients with healthcare-associated infections (HAI) [1,2,3,4,5]. Invasive candidiasis (IC) accounts for over 95% of these infections, with *Candida albicans*, *Candida glabrata*, *Candida tropicalis*, *Candida parapsilosis*, *Candida krusei*, and, in certain regions, *Candida auris* being the primary culprits. The Latin American region witnesses a significant burden of IC, with incidence rates ranging from 0.74 to 6.0 per 1000 hospital admissions and alarmingly high mortality rates ranging from 30% to 76% [6,7,8,9]. Notably, admission to intensive care units (ICUs) significantly escalates the risk, with at least half of ICU patients succumbing to this diagnosis [10]. The surge in infections by these Candida species is attributed to multiple factors, encompassing sustained exposure to antifungal agents, the utilization of catheters in hospitalized patients, underlying malignancies, advancing age, and geographic distribution [11]. Furthermore, the development of antifungal resistance and the capacity to elude host immunity add to the intricate challenge of managing candidiasis in clinical practice [12,13].

*Candida tropicalis* has emerged as one of the most important non-albicans Candida spp., due to its high incidence in systemic candidiasis and greater resistance to commonly used antifungals [14,15]. This pathogen not only possesses the capability to infiltrate vital organs [16] but is also linked to elevated mortality rates when compared to *C. albicans* and other non-albicans Candida species. *Candida tropicalis* exhibits a propensity for dissemination, particularly among neutropenic individuals and those grappling with malignancies, notably in cases demanding prolonged catheterization, extensive use of broad-spectrum antibiotics, or underlying cancer [17]. Recent increases in antifungal resistance have been associated with mutations in the *ERG11* gene, encoding ergosterol-synthase, and the overexpression of the transcriptional regulator *UPC2* [18,19]. Considering these developments, *Candida tropicalis* has emerged as a prominent etiological agent in IC, notably in Latin America and certain Asian countries, further accentuating its clinical significance [20,21].

Regrettably, the three principal antifungal drug classes (polyenes, echinocandins, and azoles) have been unable to stem the surge in life-threatening fungal infections observed over the past few decades [22]. In response to this pressing clinical challenge, compounds derived from botanical sources have gained traction as a promising alternative [23,24,25]. In our previous investigations, we established that isoespintanol (ISO), a monoterpene derived from *Oxandra xylopioides*, exhibits potent antifungal properties against *Candida tropicalis* [26,27]. Our findings elucidated ISO’s ability to induce intracellular reactive oxygen species, disrupt cell membranes, and eradicate mature biofilms as part of its multifaceted antifungal action.

However, the precise mechanism by which ISO orchestrates genetic dysregulation in *Candida tropicalis* remains an enigma. Herein, we propose that ISO plays a pivotal role in reshaping the transcriptional landscape of *Candida tropicalis*. Our primary objective is to decipher the differential gene expression patterns underlying processes such as membrane biogenesis and biofilm formation, among other critical biological pathways impacted by ISO. By delving into the potential targets of this antifungal biomolecule and their distinctive gene expression profiles, our research offers essential insights. These insights are poised to unlock novel therapeutic avenues for the treatment of candidiasis, presenting a holistic understanding of the genetic perturbations provoked by ISO.

Our study endeavors to provide a perspective on potential therapeutic targets in the battle against candidiasis. Our transcriptomics-based approach offers a comprehensive view of the biological behavior of these pathogenic yeasts and their genetic adaptations when exposed to ISO. These findings hold the promise of paving the way for innovative therapies to combat these resilient pathogens within the realm of medical practice.

## 2. Materials and Methods 

### 2.1. Isolation and Identification of Isoespintanol 

Isoespintanol (ISO) was isolated from “*yaya prieta*” leaves (*Oxandra xylopioides*) collected in October 2019. The specimen was located at coordinates 08°48′17″ north latitude and 75°42′07″ west longitude in the Municipality of Montería, Department of Córdoba, Colombia. A herbarium specimen (collection number JAUM 037849) was deposited in the Joaquín Antonio Uribe Botanical Garden in Medellín, Colombia. To isolate ISO, 5 g of petroleum benzine extract was subjected to hydrodistillation and successive crystallizations with *n*-hexane, following a modified version of the methodology described in [28]. This process yielded 1.2 g of purified ISO. The purity of ISO was confirmed using gas chromatography coupled to a Thermo Scientific model Trace 1310 mass spectrometer equipped with an AB-5MS column (30 m × 0.25 mm id × 0.25 µm). The temperature gradient system initiated at 80 °C for 10 min and increased to 200 °C at a rate of 10 °C/min. Subsequently, the temperature was raised to 240 °C at 4 °C/min and finally to 290 °C for 10 min at 10 °C/min. The injection was split less, with a volume of 1 µL. Mass spectrometry was conducted in electron impact ionization mode at 70 eV, and ion detection was performed in positive full-scan mode. The structure of ISO was elucidated through a combination of spectroscopic techniques, including ^1^H-NMR, ^13^C-NMR, DEPT, ^1^H-^1^H COSY, HMQC, and HMBC spectra. These analyses were performed on a 400 MHz Bruker Advance DRX spectrometer using deuterated chloroform (CDCl_3_) as the solvent.

### 2.2. Yeast Strain and Culture Conditions

The yeast strain utilized in this study, identified as *Candida tropicalis* (CLI007), was originally isolated from a blood culture sample obtained from a hospitalized patient at Salud Social S.A.S. in Sincelejo, Colombia. Standard methods for yeast identification, including Vitek 2 Compact, Biomerieux SA, Y.S.T. Vitek 2 Card, and AST-YS08 Vitek 2 Card (Ref 420739), were employed to initially identify the yeast strain. Further confirmation of its identification was established through a comprehensive genome-wide taxonomic study, as detailed in Contreras [26]. To maintain the yeast cultures, Sabouraud Dextrose Agar (SDA) and BBL CHROMagar Candida media were employed. Prior to conducting the experiments, a yeast suspension was meticulously adjusted to a concentration of 10^7^ colony-forming units per milliliter (CFU/mL) in phosphate-buffered saline (PBS) with a pH of 7.4. This standardized yeast suspension served as the inoculum for the subsequent assays.

### 2.3. RNA-Sequencing and Read Count Data Acquisition

Total RNA was extracted from yeast samples following a 4 h exposure to ISO at its minimum inhibitory concentration (MIC: 391.6 µg/mL) [26]. Additionally, control yeast samples without ISO treatment were processed in parallel. RNA extraction was carried out using the Trizol reagent, following the manufacturer’s standard protocol (Trizol TM Reagent, Invitrogen (Vilnius, Lithuania); Total RNA extraction protocol). The quantification of the total RNA was performed using the ribogreen colorimetric method (Invitrogen, Vilnius, Lithuania). Furthermore, the integrity of the RNA was assessed by measuring the RNA Integrity Number (RIN) through capillary electrophoresis on an Agilent 2100 bioanalyzer (Agilent Technologies, Santa Clara, CA, USA). A RIN score exceeding 7.0 was established as the threshold for acceptable RNA quality. Subsequently, the extracted RNA was employed to construct RNA-Seq libraries for sequencing, utilizing the Illumina Novaseq platform with 150 bp paired-end sequencing (Illumina-Transcriptome, TruSeq mRNA/Illumina). For the generation of RNA counts, *Bowtie 2* was utilized, aligning the sequencing data to the Ensemble GTF files corresponding to the reference genome *Candida tropicalis* (REF GCA000006335v3).

### 2.4. Bioinformatics Analyses

#### 2.4.1. Differential Gene Expression Analysis

To identify differentially expressed genes (DEGs), we utilized the read counts generated from the RNA-sequencing data. The differential gene expression analysis was conducted using the *EdgeR* package within the RStudio environment. The dataset was submitted to the analysis pipeline to assess the directionality of gene expression changes between two stages: Stage A, which comprised control/untreated *Candida tropicalis* samples, and Stage B, which included ISO-treated *Candida tropicalis* samples. We considered genes with a log2-fold change (Log2FC) greater than 1 and a false discovery rate (FDR) less than 0.05 as significant DEGs. DEGs with positive Log2-fold change and significant *p*-values were categorized as exhibiting increased expression (UP), while DEGs with negative Log2-fold change represented decreased expression (DN). Genes with *p*-values exceeding 0.05 were categorized as showing no significant change in expression between the two stages (NC).

#### 2.4.2. Functional Enrichment Analysis for DEGs

Functional enrichment analysis was conducted on the set of 186 differentially expressed genes (DEGs) identified in the *Candida tropicalis* dataset. This analysis was performed using gene set enrichment analysis (GSEA), encompassing all Gene Ontology (GO) terms. To ensure robustness, GO terms with values exceeding 0.05 were filtered out from the analysis. Subsequently, the resulting *p*-values underwent adjustment for multiple hypothesis testing using the Benjamini and Hochberg approach [29]. For the assessment of functional enrichment, two distinct gene sets were examined: the upregulated genes (UP) and the downregulated genes (DN). The background universe for these analyses was defined as the total number of DEGs. To facilitate data analysis and visualization, we employed ShinyGO 0.77 [30], based on the *Candida tropicalis* genome derived from STRING-db. This resource provided comprehensive insights into the functional enrichment of GO terms (http://bioinformatics.sdstate.edu/go, accessed on 10 May 2023).

### 2.5. Computational Prediction of DEGs and Primer Design

To enhance the functional annotation of the transcriptome, we obtained an orthologous gene pair list between *Candida tropicalis* and *Candida albicans* from the Ensembl database. This was achieved through the utilization of the BioMart package, facilitated by R scripts. From the pool of 186 differentially expressed genes (DEGs), we refined our selection to focus on genes enriched in key biological functions, namely mitochondria, cell wall, lipid metabolism, and general metabolism. For data visualization, we employed the RStudio environment to create plots featuring the selected genes. Subsequently, we designed primers for specific genes of interest, namely ERG6 (upregulated, associated with steroid metabolism), KRE1 (upregulated, linked to cell wall processes), and CTRG_03786 (downregulated, implicated in cell wall functions). These primers were designed using PrimerQuest (IDT, Integrated DNA Technologies, MI, USA, https://www.idtdna.com/pag-es/tools/primerquest?returnurl=%2FPrimerQuest%2FHome%2FIndex accessed on 16 June 2023). For primer details, please refer to Appendix A.

### 2.6. Gene Expression Assessment by Quantitative RT-PCR

For experimental validation, RNA isolation was carried out using the TRIzol^®^ reagent, following the manufacturer’s instructions precisely (Trizol TM Reagent, Invitrogen, Vilnius, Lithuania; Total RNA extraction protocol), as previously detailed in Section 2.3. For gene expression analysis, cDNA synthesis was conducted using SuperScript III (Invitrogen Cat No. 12574026) following the manufacturer’s recommended protocols. Quantitative RT-PCR was performed using the StepOnePlus system (Applied Biosystems, Vilnius, Lithuania) with Power Up SYBR Master Mix (Applied Biosystems). This approach allowed us to determine the relative expression levels of mRNA transcripts for three target genes: ERG6, KRE1, and CTRG_03786. The housekeeping gene ACTIN-1 served as the reference for normalization. Each reaction mixture consisted of 100 ng of cDNA, 2X master mix, and 500 picomolar of each primer, resulting in a final volume of 10 μL. PCR cycling conditions encompassed an initial denaturation step at 95 °C for 2 min, followed by 40 cycles of 15 s at 95 °C, 15 s at 62 °C, and 60 s at 72 °C. The primer sequences for ERG6, KRE1, CTRG_03786, and ACTIN-1 [31] are detailed in Appendix A. Gene expression analysis was carried out employing the 2^(−ΔΔCT)^ method of relative quantification [32].

### 2.7. Determination of Total Ergosterol Content

The quantification of the total ergosterol content in *C. tropicalis* isolates treated with ISO was conducted in accordance with the protocol outlined in [33], with some modifications. Initially, the cells were exposed to ISO at their respective minimum inhibitory concentrations (MICs) and incubated at 35 °C for a duration of 3 h. Subsequently, the treated cells were subjected to centrifugation and washed with phosphate-buffered saline (PBS). To initiate ergosterol extraction, a wet weight of 0.5 g of cells was mixed with PBS. Saponification was achieved by adding 4 mL of a freshly prepared 30% (*w*/*v*) methanolic KOH solution and 8 mL of absolute ethanol. This mixture was maintained at 80 °C for a period of 1 h. The ensuing mixture was extracted using petroleum ether and then washed with a saturated NaCl solution. Following the extraction, the samples were concentrated under vacuum conditions at 60 °C. The resulting residue was dissolved in 0.5 mL of methanol and subsequently filtered through a 0.45 µm micromembrane. The quantification of ergosterol content was determined by comparing the peak areas of the samples against a standard curve generated from ergosterol (95% Sigma-Aldrich, St. Louis, MO, USA). The concentrations used in the standard curve spanned 1, 10, 50, 100, 250, 500, 750, and 1000 mg/L of ergosterol. Ergosterol content analysis was performed using a UHPLC Ultimate 3000 system (Thermo Scientific, Waltham, MA, USA) equipped with a diode array detector (DAD). A C18 Hypersil Gold column (150 mm × 4.6 mm, 5 µm) was maintained at 30 °C. Each injection had a volume of 30 µL, and the mobile phase consisted of methanol/water (97/3, 100% HPLC grade) at a flow rate of 0.6 mL/min. All measurements were conducted in triplicate, and ergosterol was detected at a wavelength of 205 nm. 

The ergosterol inhibition ratio was calculated as follows:

Ergosterol Inhibition Ratio = (1 − Ergosterol Content of Treated Cells/Ergosterol Content of Untreated Cells) × 100%.

## 3. Results

### 3.1. Isolation and Purification of Isoespintanol Molecules

Isoespintanol (ISO) in the amount of 1.2 g was isolated and purified to a crystalline amorphous solid with a purity exceeding 99%. The high purity was verified using gas chromatography-mass spectrometry (GC-MS). Structural identification of ISO was accomplished through various spectroscopic techniques, including ^1^H-NMR, ^13^C-NMR, DEPT, ^1^H-^1^H COSY, HMQC, and HMBC. This rigorous analysis led to the unequivocal proposal of the structure of 2,5-dimethoxy-3-hydroxy-*p*-cymene isoespintanol, as illustrated in Figure 1. Furthermore, electron impact mass spectrometry (EI-MS) provided essential data, with the observed molecular ion at *m*/*z* 210 (49%) and prominent fragment ions at *m/z* 195 (100%), 180, 165, 150, 135, and 91. Detailed nuclear magnetic resonance (NMR) spectra were recorded. In the ^1^H-NMR spectrum (400 MHz, CDCl_3_), the chemical shifts were observed at δ 6.22 (singlet, 1H, H6), δ 5.85 (singlet, 1H, HO-3), δ 3.77 (singlet, 3H, H12), δ 3.76 (singlet, 3H, H11), δ 3.52 (heptet, J = 7.1 Hz, 1H, H8), δ 2.29 (singlet, 3H, H7), and δ 1.33 (doublet, J = 7.1 Hz, 6H, H9-H10). In the ^13^C-NMR spectrum (100 MHz, CDCl_3_), the carbon chemical shifts were observed at δ 154.3 (C5), δ 147.4 (C3), δ 139.7 (C2), δ 126.8 (C1), δ 120.4 (C4), δ 104.4 (C6), δ 24.6 (C8), δ 60.8 (C11), δ 55.7 (C12), δ 20.6 (C9, C10), and δ 15.8 (C7).

### 3.2. RNA Preparation, Library Construction, and Sequence Analysis

The results of RNA extraction from *Candida tropicalis*, both with ISO treatment and without treatment (control), are summarized in Appendix A. Specifically, we obtained a total RNA yield of 0.797 µg for ISO-treated samples and 0.785 µg for untreated control cells. For a detailed overview of the sequencing statistics and transcriptome data, including the mapping of reads to the *Candida tropicalis* reference genome (REF GCA000006335v3), please refer to Appendix A.

Transcriptomes obtained by RNA-seq were mapped on the reference genome *Candida_tropicalis* REF GCA000006335v3.

### 3.3. Analysis of the Transcriptional Profile of Differentially Expressed Genes

Our findings reveal distinct patterns of gene expression between yeast exposed to ISO and untreated yeast. A comprehensive analysis identified a total of 186 differentially expressed genes. Among these, 159 genes displayed induction in their expression levels, while 27 genes exhibited repression (Figure 2).

Figure 3 presents the key DEGs that are upregulated and downregulated in response to ISO treatment in the *C. tropicalis* dataset. Noteworthy among the upregulated genes is the SET domain methyltransferase protein superfamily, encompassing proteins known for their histone methylation activity at lysine residues. Histone methylation is a pivotal process involved in chromatin regulation and gene expression [34]. Additionally, the upregulated genes encompass components of transmembrane transport systems, particularly those associated with the allenoate transport system [35]. Furthermore, genes related to the 3β-HSD enzyme system, crucial for the biosynthesis of various classes of steroid hormones, are observed [36,37]. The dataset also includes hydrolase enzyme genes involved in sphingolipid metabolism reactions [38,39], as well as genes contributing to the structural integrity of the ribosome.

#### 3.3.1. Stress Response, Metabolism, and Mitochondrial Functional Processes Are Enriched in Upregulated Genes in *C. tropicalis* Treated with ISO

In response to ISO treatment, upregulated genes in *C. tropicalis* were found to be associated with critical biological processes. Notably, these upregulated genes included ATP-dependent proteases of the Clpa/b family and the DnaJ domain (DnaJ/Hsp40: heat shock protein 40), which serve as essential chaperones involved in translation, protein folding, and protein translocation [40,41]. This suggests that ISO may impact protein synthesis in this yeast. Furthermore, the expression of the STI1 domain, known for its presence in the DNA damage response protein Rad23 [42], was notable. This domain has a role in recognizing DNA damage, indicating that ISO’s antifungal action might involve DNA damage to these pathogens. The upregulated genes also included those from the Aldo-Keto Reductase (AKR) superfamily, catalyzing redox transformations essential for biosynthesis, intermediary metabolism, and detoxification [43]. Additionally, overexpression of zinc-binding dehydrogenase family proteins, involved in oxidoreductase and catalytic activities, was observed, suggesting their role in responding to oxidative stress. Moreover, proteins related to the cellular response to organic substances, such as dioxygenases catalyzing critical reactions in aerobic microbial degradation of aromatic compounds [44], were upregulated. Figure 4 illustrates the overexpression of genes associated with ergosterol biosynthesis, mitochondrial function, signaling pathways, and carbohydrate metabolism.

#### 3.3.2. Upregulation of Specific Steroid and Cell Metabolism Genes Is an Effect of *C. tropicalis* Treated with ISO

Treatment of *Candida tropicalis* with ISO at its minimum inhibitory concentration (MIC) led to the notable upregulation of specific genes involved in ergosterol biosynthesis, including ERG1, ERG6, ERG2, and ERG26 (Figure 5). Ergosterol is a fundamental component of the plasma membrane, and the increased expression of these genes suggests that ISO might impact its synthesis. Additionally, our results indicate the induction of genes associated with metabolism and the upregulation of Vip1, an inositol polyphosphate kinase known for its role as a signaling molecule. Vip1 regulates various essential biophysical processes, including autophagy and pathogenicity in *C. albicans* [45].

#### 3.3.3. ISO-Induced Dysregulation of Mitochondrial and Cell Wall Genes in *C. tropicalis*

We also deepen insight into the genes that undergo dysregulation in response to ISO treatment, particularly those associated with mitochondrial function and cell wall pathways (Figure 6). Noteworthy among these genes is KRE1, which is involved in the synthesis of β-glucan in the cell wall of *C. albicans* [46]. Additionally, the upregulation of the btgC gene, encoding a glucan endo-1,3-β-glucosidase, suggests potential involvement in β-glucan degradation. Furthermore, the overexpression of Mid2, cell surface sensors in cell wall integrity signaling [47], and calcium-activated chloride channels (CaCC), essential for cell physiology [48], was observed in *C. tropicalis*. ISO treatment also induced the overexpression of HSP82 heat shock proteins, an essential gene family in yeast cells [49]. Additionally, enzymes such as formate dehydrogenase (FDH), crucial metalloenzymes catalyzing the reversible conversion of formate into carbon dioxide [50], were upregulated. These enzymes can utilize various electron donors, including ferredoxin, NAD, NADP, quinones, or F420 [51], playing a vital role in mitochondrial metabolism.

Furthermore, the upregulation of ATP-dependent-(S)-NAD(P)H-hydrate dehydratase, catalyzing the dehydration of the S-form of NAD(P)HX at the expense of ATP, converting it to ADP, was also observed.

#### 3.3.4. Gene Expression Validation Using qPCR

Experimental validation was conducted using quantitative PCR (qPCR) to confirm the differential expression of specific genes in *C. tropicalis* treated with ISO, as depicted in Figure 7. Notably, the CTRG_03786 and ERG6 genes exhibited 1.14- and 1.8-fold higher expression, respectively, compared to the housekeeping gene in the case group. In contrast, the KRE1 gene demonstrated an eightfold higher expression in yeasts treated with isoespintanol compared to ACTIN-1. These qPCR results align with the findings obtained from the biostatistical analysis of RNA-seq.

#### 3.3.5. Methylation-Related Genes Are Downregulated in *C. tropicalis* after ISO Treatment

The downregulation of genes associated with methylation in response to ISO treatment, as illustrated in Figure 8, highlights an interesting facet of the antifungal activity of ISO against *C. tropicalis*. One of the downregulated genes, TRM7, encodes the tRNA methyltransferase TRM7, which is responsible for introducing modifications to transfer RNA (tRNA) [52,53]. These modifications are crucial for tRNA functionality in processes such as translation. The downregulation of TRM7 implies that ISO treatment may interfere with tRNA modifications, potentially affecting protein synthesis and cellular metabolism in *C. tropicalis.* Similarly, the downregulation of the TRM2 gene, which encodes tRNA(m5U54) methyltransferase, indicates ISO’s impact on the formation of modified nucleosides within tRNA [54]. These modifications are essential for tRNA stability and functionality in protein synthesis. The disruption of this process could further contribute to the inhibition of protein synthesis in the yeast cells.

One of the most intriguing findings is the deregulation of the DOT1 gene, which encodes a histone methyltransferase targeting nucleosomal H3-Lys79. Histone modifications play a crucial role in chromatin remodeling and gene expression regulation [55,56,57,58]. The involvement of DOT1 in critical cellular processes, such as the DNA damage checkpoint, nucleotide excision repair, recombination, and chromatin silencing at telomeres, underscores the multifaceted impact of ISO on *C. tropicalis.* The fact that DOT1 is primarily located in the nucleus suggests that ISO treatment may disrupt chromatin structure and gene regulation in the yeast [55,56,57,58]. This could have downstream effects on DNA repair mechanisms and overall cellular integrity.

In summary, the downregulation of methylation-related genes in response to ISO treatment provides insight into the complex molecular mechanisms through which ISO exerts its antifungal activity. It highlights the potential disruption of essential cellular processes, including translation, nucleotide metabolism, and chromatin regulation, ultimately contributing to the inhibition of *C. tropicalis* growth and survival. Further research is needed to elucidate the precise molecular interactions underlying these effects and their implications for antifungal therapy.

#### 3.3.6. Downregulated Genes Are Enriched in Cell Membrane Roles and S-Adenosyl-L-Methionine (SAM) Pathways in Response to ISO Treatment 

We analyzed the functional enrichment pathways of the downregulated genes. In this analysis, in addition to the previously mentioned downregulated genes in *C. tropicalis* exposed to ISO treatment, several genes exhibited downregulation, shedding light on the multifaceted impact of ISO on the yeast’s transcriptome. These downregulated genes encompass membrane proteins and SAM-related processes, each with unique implications for the cellular functions and metabolism of the yeast. Among the downregulated genes, members of the membrane protein family, including mug84GPR1/FUN34/YaaH and mug86GPR1/FUN34/Satp, drew attention. These genes encode membrane proteins that are involved in acetate permease activity [59], a process responsible for transporting acetate across the fungal cell membrane. The downregulation of these genes implies that ISO treatment may disrupt acetate transport in *C. tropicalis*, potentially affecting various metabolic pathways. Furthermore, these membrane proteins are known to play a role in the regulation of morphogenesis and hyphal formation in *C. albicans* [60,61]. Consequently, their downregulation by ISO treatment may contribute to the inhibition of hyphal growth and morphological changes observed in *C. tropicalis*. Moreover, the downregulation of S-adenosyl-L-methionine (SAM) was observed. SAM is a critical nucleoside that serves as a central molecule in various cellular processes. It acts as a methyl donor for trans-methylation reactions, playing essential roles in epigenetic modifications, gene regulation, and protein methylation [62,63]. Additionally, SAM is involved in trans-sulfuration and trans-propylamine reactions, contributing to the synthesis of crucial molecules such as cysteine and polyamines. The downregulation of SAM in response to ISO treatment suggests that this monoterpene disrupts these essential metabolic pathways. Consequently, this disruption may affect gene regulation, protein function, and the overall cellular physiology of *C. tropicalis.*

In summary, the downregulation of specific membrane protein genes and SAM-related processes highlights the diverse and profound effects of ISO on the yeast’s cellular processes. ISO’s impact extends beyond the previously discussed deregulated genes and encompasses disruptions in acetate transport, morphogenesis regulation, and critical metabolic pathways like trans-methylation and trans-sulfuration. These findings underscore the intricate and multifaceted nature of ISO’s antifungal mechanism, providing valuable targets for further exploration into the molecular mechanisms underlying ISO’s antifungal activity and its potential as a therapeutic agent against *C. tropicalis* infections (Figure 9).

#### 3.3.7. Overexpression of Zinc-Regulated Transcription Factors (TFs) in *Candida tropicalis* Induced by ISO Treatment

In this study, we observed a notable overexpression of genes encoding transcription factors (TFs) in *C. tropicalis* in response to ISO treatment, shedding light on the intricate transcriptional changes elicited by this monoterpene. The upregulation of these TFs indicates their pivotal role in orchestrating various cellular responses and adaptations to ISO exposure. The observed overexpression of zinc-regulated TFs in *C. tropicalis* subjected to ISO treatment sheds light on the intricate transcriptional landscape influenced by this monoterpene. The upregulation of GAL4-like Zn, a DNA-binding TF, suggests its involvement in regulating the transcription of specific genes. GAL4-like Zn is known for its role in recognizing the palindromic DNA sequence of galactose metabolizing enzyme genes [64]. The overexpression of this TF may signify a shift in metabolic processes in response to ISO treatment, potentially influencing galactose metabolism or related pathways. Another induced TF is RRN11, a DNA-binding TF for RNA polymerase I, it plays a crucial role in the transcription of large nuclear rRNA transcripts [65]. Its overexpression suggests potential alterations in rRNA transcription in *C. tropicalis* following ISO exposure, which may impact ribosomal biogenesis and protein synthesis. The upregulated GAL4-like Zn TF hints at potential alterations in metabolic pathways, possibly affecting galactose metabolism or related processes [64]. Concurrently, the upregulation of RRN11 TF points toward changes in rRNA transcription, impacting ribosomal biogenesis and protein synthesis [65]. Moreover, the overexpression of RPN4 TF suggests ISO-induced stress responses, potentially affecting protein turnover and DNA repair mechanisms [66,67]. The upregulated USV1 TF implies shifts in respiratory pathways, impacting energy metabolism and cellular respiration [68,69]. Additionally, the upregulation of ZPR1 suggests potential disruptions in the cell cycle progression [70]. ELA1 overexpression indicates ISO-induced DNA damage responses [71]. Furthermore, the overexpression of UPC2 TF implies ISO’s impact on ergosterol biosynthesis pathways, influencing membrane composition and integrity [72,73]. Lastly, the overexpression of genes related to environmental stress responses, such as STP4, suggests that ISO triggers stress-related adaptations in *C. tropicalis* [74] (Figure 10).

Overall, the overexpression of these zinc-regulated TFs underscores the complex and multifaceted transcriptional responses of *C. tropicalis* to ISO treatment. These TFs play pivotal roles in regulating various cellular processes, including metabolism, stress responses, DNA damage repair, and cell cycle progression. Understanding the regulatory networks governed by these TFs is crucial for unraveling the mechanisms underlying ISO’s antifungal activity and its potential as a therapeutic agent.

### 3.4. Disruption of Fungal Membrane Integrity: Ergosterol Content Analysis Reveals ISO’s Antifungal Mechanism

The analysis of ergosterol content was conducted using UHPLC-DAD, revealing pertinent insights into the impact of ISO treatment on *C. tropicalis*. The retention time for ergosterol, as determined by UHPLC-DAD analysis, was consistently observed at approximately 10.8 min (Figure 11). The ergosterol content was quantified in both control (T1) and ISO-treated (T2) samples following a 3 h incubation period. Notably, the ergosterol content in the control samples (T1) was measured at 28.40 ± 1.48 mg/L, while ISO-treated samples (T2) exhibited a notably reduced ergosterol content of 15.58 ± 0.37 mg/L (*p* < 0.05). Furthermore, the standard curve employed in the analysis demonstrated excellent linearity, as evidenced by an R^2^ value of 0.9965. These findings unequivocally indicate that ISO treatment resulted in a substantial reduction in ergosterol content in the membrane of *C. tropicalis*, underscoring the pronounced effect of ISO on the fungal membrane composition. The significant reduction in ergosterol content in ISO-treated samples compared to control samples is a compelling finding. Ergosterol is essential for maintaining the integrity and fluidity of the fungal cell membrane. Therefore, a decrease in ergosterol levels can lead to membrane dysfunction, compromising the ability of *C. tropicalis* to adapt and survive in different environments. This reduction in ergosterol content is consistent with the observed upregulation of genes associated with ergosterol biosynthesis in the transcriptomic analysis, suggesting that ISO may disrupt the synthesis of this critical membrane component.

The linearity of the standard curve, as indicated by an R^2^ value of 0.9965, further validates the accuracy of the ergosterol quantification method, strengthening the credibility of the results. These findings collectively emphasize the pronounced effect of ISO on the fungal membrane composition, which is crucial for the pathogenicity and survival of *C. tropicalis*. Understanding the impact of ISO on ergosterol content provides valuable information for future research and the development of antifungal therapies targeting membrane integrity and function.

## 4. Discussion

*Candida tropicalis* has gained prominence within the Candida genus due to its rising incidence, drug resistance, and heightened mortality rates, particularly among immunocompromised individuals [14,16,18,75,76]. Despite the effectiveness of existing antifungal drugs, the emergence of drug-resistant Candida spp. presents a significant therapeutic challenge [77]. Furthermore, the limited treatment options and potential toxicity associated with current therapies necessitate the exploration of novel antifungal compounds, with natural plant-based compounds emerging as promising alternatives [78,79,80,81,82,83]. In prior studies, we established the efficacy of isoespintanol (ISO) against clinical Candida spp. isolates, including *C. tropicalis*, and elucidated its molecular targets [26,27,84]. In this study, we employ a transcriptomic approach to delineate gene expression profiles, providing insights into the intricate mechanisms underlying ISO’s antifungal activity.

Ergosterol, an essential component of fungal cell membranes, governs membrane fluidity, permeability, and the activity of membrane-associated proteins [85]. It is distributed across various cellular components, including the cell membrane, intracellular endomembranes, and mitochondrial membranes [86]. Because ergosterol is absent in mammals, it serves as an attractive target for antifungal agents. These lipids house numerous biologically essential proteins involved in signaling, stress responses, nutrient transport, and other vital processes [77]. Ergosterol biosynthesis is a complex process regulated by numerous ERG genes [87]. Our findings reveal the upregulation of ERG1, ERG6, ERG2, and ERG26 genes, all involved in ergosterol biosynthesis. The precise mechanism responsible for the global upregulation of ERG genes in response to azoles remains unclear. However, it has been postulated that disruptions in ergosterol or other sterol pathway components lead to widespread ERG gene expression increases [88]. Alternatively, the accumulation of early substrates or toxic sterol byproducts may trigger ERG expression [33,89]. This study demonstrated a reduction in *C. tropicalis* ergosterol content in response to ISO treatment, leading to an increase in ERG gene expression. Inhibition of the ergosterol synthesis pathway results in an accumulation of toxic sterols, which, coupled with the reduction in ergosterol, damages the cell membrane’s integrity, inhibiting fungal cell growth [90]. These results align with previous findings where *C. albicans* treated with azoles exhibited similar responses [91,92]. Notably, the ERG6 gene and its protein products may serve as valuable antifungal targets for a new generation of ergosterol biosynthesis inhibitors, such as ISO. This gene is particularly interesting because it encodes sterol C-24 methyltransferase, which is not involved in cholesterol biosynthesis in human cells. Furthermore, it has been reported that ERG6 deletion leads to hypersensitivity to several metabolic inhibitors and an inability to import tryptophan or utilize respiratory energy sources. Hence, the use of inhibitors that target the ERG6 gene product could render pathogenic yeast hypersensitive to currently known antifungal agents, as well as novel compounds [93,94]. Furthermore, the regulation of ergosterol synthesis involves overlapping mechanisms, including transcriptional expression, feedback inhibition of enzymes, and changes in subcellular localization [85]. Considering the pivotal physiological role of ergosterol in these pathogenic yeasts [95] and ISO’s impact on ergosterol biosynthesis and associated transcription factors, ISO emerges as a promising compound in the battle against these pathogens.

It is well known that fungal cell walls are critical for viability, morphogenesis, and pathogenesis [96]. *Candida albicans*’ cell wall comprises an inner skeletal layer rich in β-(1,3)- and β-(1,6)-glucan and chitin, along with an outer fibrillar layer primarily composed of highly mannosylated cell wall proteins [97,98]. Given that the cell wall plays a crucial role in maintaining cellular osmotic integrity, any damage to it triggers various responses [99], previous studies have shown that inhibiting β-(1,3)-glucan synthesis leads to compensatory chitin synthesis and alterations in wall macromolecule arrangements in *C. albicans.* Similarly, *C. tropicalis* has been shown to activate a chitin compensatory response upon exposure to D-limonene [98] and ISO [84].

Of particular interest is the upregulation of the KRE1 gene, which is involved in β-glucan synthesis and assembly in yeast cell walls like *Saccharomyces cerevisiae* and *C. albicans.* Mutations in this gene are associated with abnormal (1,6)-β-glucan production in yeast cell walls [46,100,101]. This suggests that ISO may indeed alter the cell wall structure in *C. tropicalis.* As cell wall β-glucans are common in fungi and absent in mammals, enzymes involved in β-glucan biosynthesis are potential targets for specific antifungal agents [96,102].

Monoterpenes have been reported to interact with various intracellular targets, including DNA and RNA, leading to disruptions in protein synthesis, metabolic processes, biosynthetic pathways, mitochondrial membrane potential, and more [103,104]. Our results indicate that ISO can penetrate *C. tropicalis* cells, interacting with intracellular structures, thereby disrupting the pathogens’ normal metabolic processes, enzymatic activity, mitochondrial function, and signaling pathways. Notably, we observe the upregulation of genes related to protein translation and folding, suggesting that protein synthesis may be a critical target in ISO’s antifungal activity. Furthermore, the upregulation of the STI1 domain, known for its role in various cellular processes, including the transfer of hydrophobic substrates and the DNA damage response, suggests that ISO’s antifungal action also involves DNA damage in *C. tropicalis*. Our results are in accordance with previous studies in *C. tropicalis* and *C. parapsilosis*, where terpenes like D-limonene have exhibited multiple action targets [24,25,103].

Additionally, we note the downregulation of methylation-related genes in *C. tropicalis* following ISO treatment. Genes such as TRM7, TRM2, and DOT1, known for their roles in tRNA modification and histone methylation, were downregulated [52,53,54,55,56,57,58]. This phenomenon suggests that ISO may alter key methylation processes within yeast cells, ultimately affecting various vital cellular functions. Moreover, genes involved in acetate permease activity and the expression of S-adenosyl-L-methionine (SAM), a crucial nucleoside serving as a methyl donor in various processes, including choline, carnitine, creatine, DNA, and protein methylation, were found to be downregulated [105]. These findings suggest that ISO’s antifungal action involves complex regulation processes, potentially including post-transcriptional events.

Interestingly, our results reveal the upregulation of genes encoding transcription factors (TFs) in response to ISO treatment. These include GAL4-like Zn, RRN11, RPN4, USV1, ZPR1, ELA1, and UPC2. These TFs play various roles, from binding to specific DNA sequences to regulating DNA repair and stress responses. This deregulation underscores the complexity of ISO’s antifungal mechanism, suggesting that it may involve multiple regulatory processes. These findings highlight the multifaceted nature of ISO’s action against *C. tropicalis*.

In summary, ISO’s antifungal activity against *C. tropicalis* employs a multi-molecular target approach, encompassing disruptions in ergosterol and protein biosynthesis, alterations in cell wall and membrane structure, perturbation of multiple cellular processes, and potentially the induction of DNA damage. Compounds with multiple molecular targets have been shown to enhance activity against multidrug-resistant strains and inhibit resistance development [106]. Consequently, ISO, by targeting various aspects of the biology of these pathogens, could serve as a versatile tool for different objectives or as a starting point for the development of new drugs, drug precursors, or adjuvants in the treatment and control of these pathogens. This approach holds promise in countering drug resistance and improving treatment outcomes against *C. tropicalis.* Furthermore, considering that some ISO targets are not present in human cells, this monoterpene could be a compelling candidate for designing, synthesizing, and developing new molecular prototypes with antifungal potential. The intricate nature of ISO’s antifungal mechanism underscores its potential as a promising antifungal agent against drug-resistant Candida species.

## 5. Conclusions

In this study, we conducted a comprehensive transcriptomic analysis of *C. tropicalis* exposed to ISO, a natural monoterpene isolated from *O. xylopioides.* Our findings not only reaffirm the antifungal properties of ISO, as previously reported, but also shed light on the intricate mechanisms underlying its antifungal activity. The complexity of ISO’s action, as revealed by our results, underscores its potential as a promising antifungal agent against *C. tropicalis,* a clinically significant pathogenic yeast known for its increasing drug resistance and mortality rates among immunocompromised individuals. These results provide valuable insights into potential clinical molecular targets within *C. tropicalis* that may be explored in the future for the development of novel treatments against this pathogenic yeast. This research contributes to our understanding of the molecular basis of ISO’s antifungal activity and opens new avenues for the development of effective therapies for combating drug-resistant Candida species in clinical settings.

## Figures and Tables

**Figure 1 jof-09-01199-f001:**
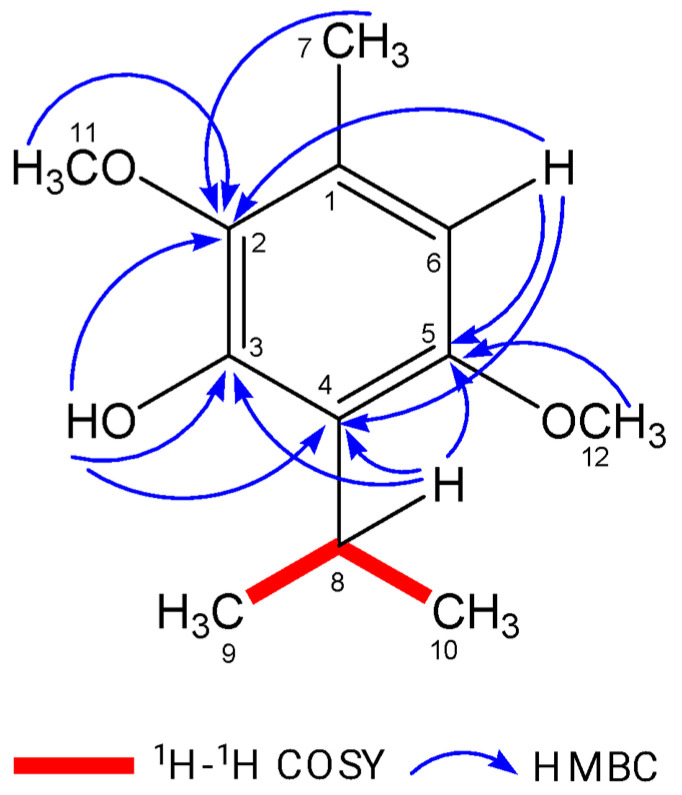
Key ^1^H–^1^H COSY and HMBC correlations of ISO.

**Figure 2 jof-09-01199-f002:**
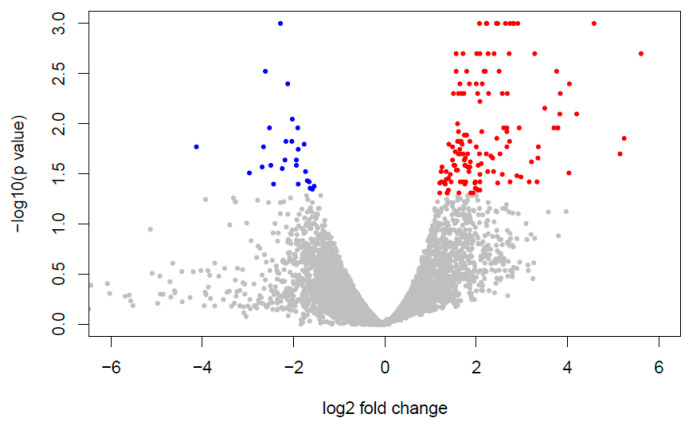
Differential gene expression analysis. Volcano plot of DEG; red (up) and blue (down) colored dots indicate the DEGs, while the gray dots represent genes without expression changes (NC) among *Candida tropicalis* and control samples.

**Figure 3 jof-09-01199-f003:**
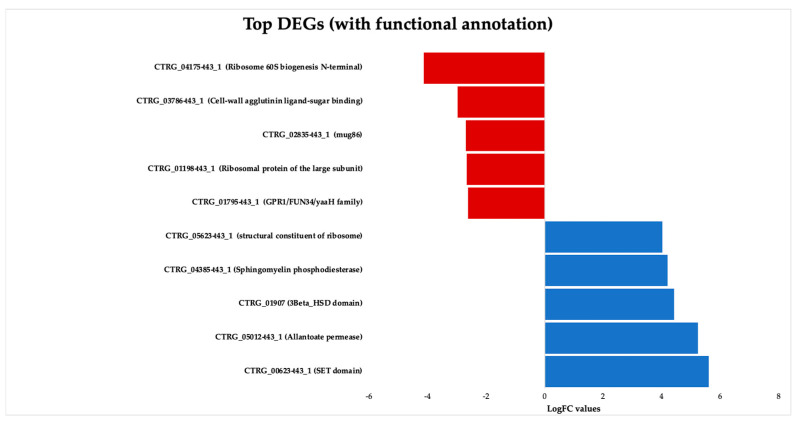
Top upregulated and downregulated genes with functional annotations in *Candida tropicalis* dataset. Bars show the top upregulated (red) and downregulated (blue) genes, according to their Log2FC values.

**Figure 4 jof-09-01199-f004:**
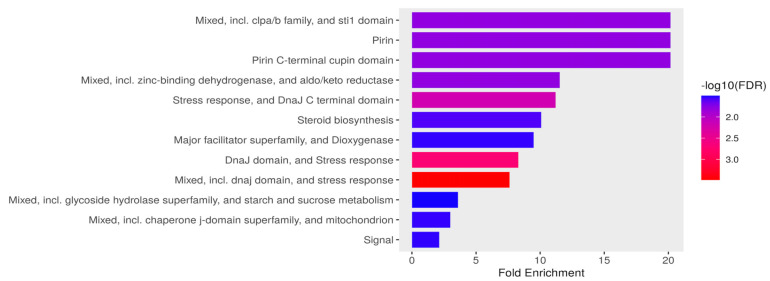
Functional enrichment of DEG in *Candida tropicalis* treatment. FDR is calculated based on a nominal *p*-value from the hypergeometric test. Fold Enrichment is defined as the percentage of differentially expressed genes belonging to a pathway, divided by the corresponding percentage in the background. FDR reports how likely the enrichment is by chance; higher values are colored on a scale of red to blue. In the x-axis, Fold Enrichment indicates how drastically genes of a certain pathway are overrepresented.

**Figure 5 jof-09-01199-f005:**
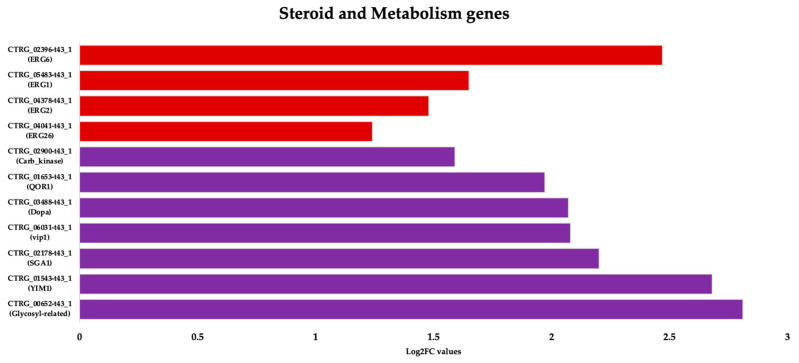
Functional annotation of upregulated DEGs in *Candida tropicalis* following ISO treatment, with focus on genes associated with steroid biosynthesis (ERG6, ERG26, ERG2, ERG1) represented by the red bars, showcasing their upregulation based on Log2FC values. Additionally, the purple bars highlight the upregulated genes involved in cell metabolism functions, reflecting their corresponding Log2FC values.

**Figure 6 jof-09-01199-f006:**
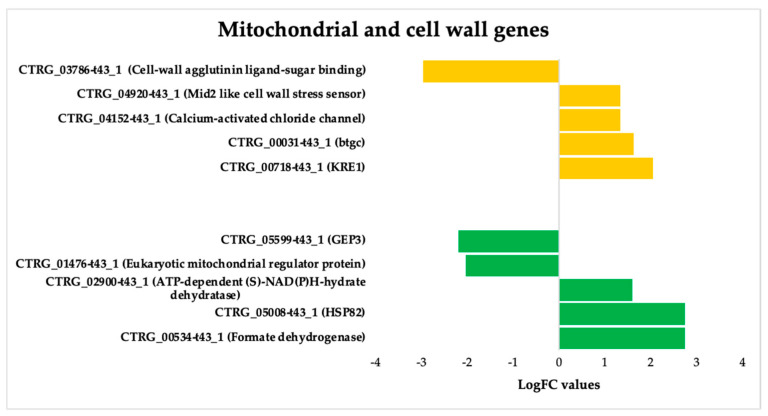
Deregulated genes in *Candida tropicalis* treatment enriched in mitochondria and cell wall processes. Green bars depict the deregulated transcripts involved in mitochondrial processes according to their Log2FC values, while cell-wall-related transcripts are depicted by yellow bars according to their corresponding Log2FC values.

**Figure 7 jof-09-01199-f007:**
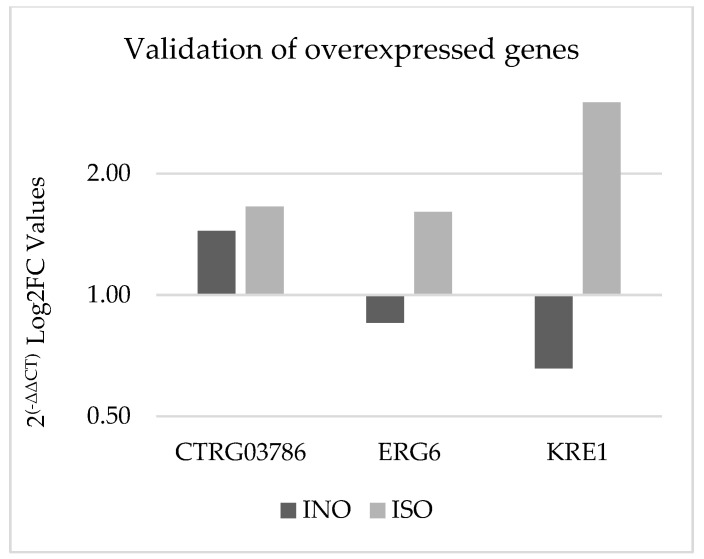
Validation of gene overexpression by qPCR using the 2^(−ΔΔCT)^ method. The dark bars represent the genetic expression of *C. tropicalis* without treatment (control group), and the light bars represent *C. tropicalis* treated with ISO. The data are presented by calculating the logarithm to base 2 (Log2FC).

**Figure 8 jof-09-01199-f008:**
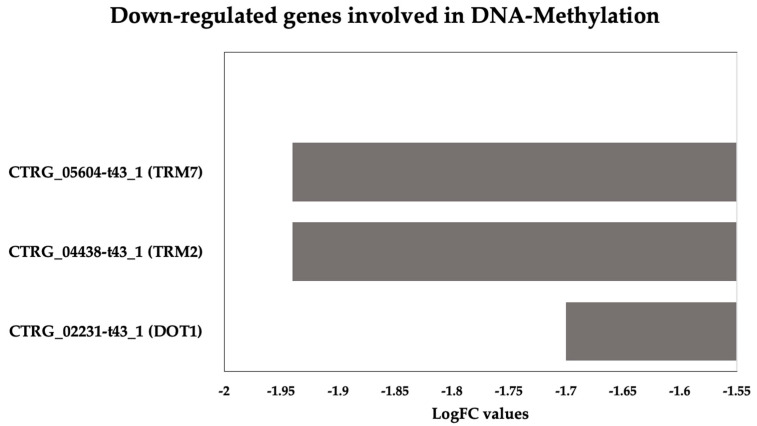
DNA methylation Down-DEGs in *Candida tropicalis* subsequent to ISO treatment. Gray bars represent the downregulated genes (TRM7, TRM2, DOT1) in accordance with their respective Log2FC values.

**Figure 9 jof-09-01199-f009:**
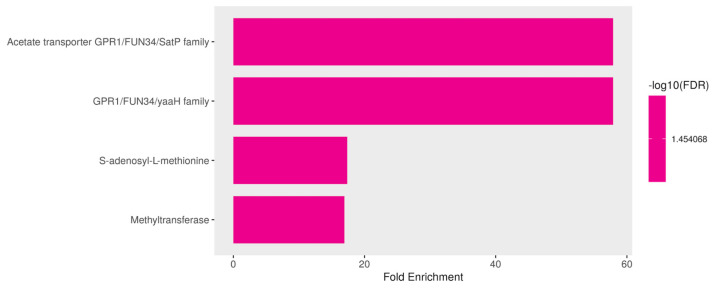
Functional enrichment of downregulated genes in *Candida tropicalis* treated with ISO. FDR is calculated based on a nominal *p*-value from the hypergeometric test. Fold Enrichment is defined as the percentage of differentially expressed genes belonging to a pathway divided by the corresponding percentage in the background. FDR reports how likely the enrichment is by chance. In the x-axis, Fold Enrichment indicates how drastically genes of a certain pathway are overrepresented.

**Figure 10 jof-09-01199-f010:**
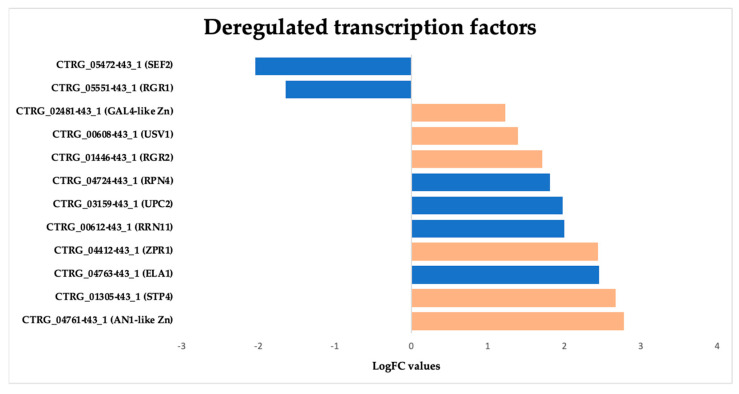
Deregulated transcription factors in *Candida tropicalis* treatment. The orange bars represent the dysregulated zinc domain transcripts, while the blue bars represent other differentially expressed transcription factors (DEG TFs) based on their Log2FC values.

**Figure 11 jof-09-01199-f011:**
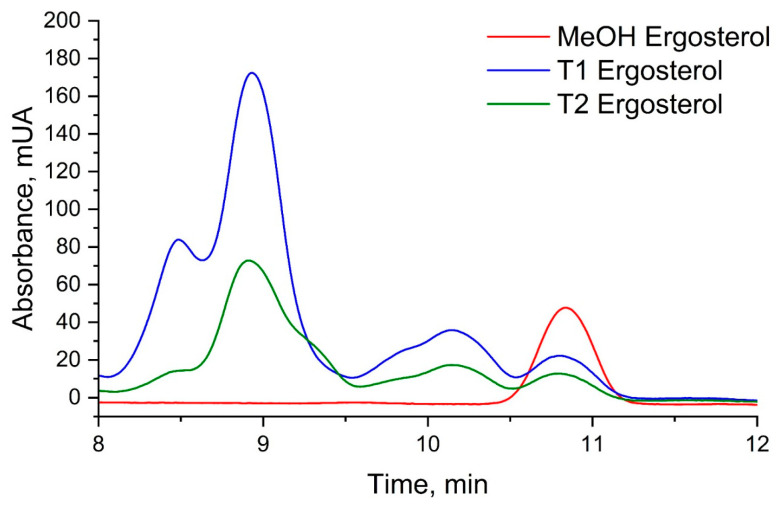
UHPLC chromatograms of ergosterol in *Candida tropicalis* with and without ISO treatment. Ergosterol extractions from the negative control and the ISO-treated group were dissolved in methanol for analysis. Each graph represents the results of three repeated experiments.

## Data Availability

The datasets used are available from https://github.com/kap8416/transcriptomicsofcandidatropicalis_isoespintanol accessed on 16 June 2023. Code and Appendix A have been deposited in https://github.com/kap8416/transcriptomicsofcandidatropicalis_isoespintanol accessed on 16 June 2023.

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
