# Peer review of "Transcriptional Reprogramming of *Candida tropicalis* in Response to Isoespintanol Treatment"

_jof, 2023, doi:10.3390/jof9121199_

Round 1

Reviewer 1 Report

Comments and Suggestions for Authors

The authors reported ˝diverse and profound effects of ISO on various yeast cellular processes˝ based on the up- and down-regulation of specific genes after ISO treatment of yeast. However, I believe that such statements are somewhat exaggerated because they are without evidence on a biochemical level. The paper generally provides a large set of data, some of which the authors tried to put into context and present as evidence of the action of ISO on yeast. The whole story, however, remains at the level of speculation and my opinion is that the statements and conclusions throughout the paper are exaggerated. As for the quality of the manuscript, the main drawback is in the figures. The number of figures is too large - most of the results in the figures are literally described in the text. Figure descriptions are poor in most cases and figures are not self-explanatory. Some specific comments are as follow:

Fig. 2 – there is no need for pie chart part – all that it shows is written in the text.

Fig. 3 – markings on diagram X -axis are not correct – e.g. gene GPR1 is mentioned two times (for two bars in the diagram), names of other down regulated genes are not gene names at all (ribosome 60S biogenesis N-terminal ?, cell wall agglutinin N-terminal sugar binding  ???). Figure title states that it is representing ˝top DEGs˝ – data shown should be in accordance with that, or figure title should be changed.    

Fig. 5 – some genes are listed by name, but diagram shows names like ˝glycosyl˝ and ˝dehydration˝ as well? That should be corrected. Figure description is unclear - what is meaning of the sentences as ˝Glycoside hydrolase superfamily, and starch and sucrose metabolism.˝?   Furthermore, in sentence ˝Purple Bars show the upregulated enriched genes according to their corresponding Log2FC values.˝ - what is the difference between upregulated and enriched?

Fig. 8 – the title and description of figure 8 is completely incomprehensible. TRM7 is not even shown in the figure.

Proteins mug84GPR1/FUN34/YaaH and mug86GPR1/FUN34/Satp are showed in the Fig. 8. as proteins involved in methylation, and then again further in the text mentioned as proteins that are involved in acetate permease activity? What is the truth?

In line 419 authors stated that S-Adenosyl-Methionin (SAM) is down-regulated. SAM is a chemical compound - how can a chemical compound be down-regulated?

It is not clear what Fig. 9 is showing? Is there functional enrichment or down regulation of these genes? Not to mention that S-Adenosyl-Methionine is not gene. Why is there color-coded FDR shown when all bars are of the same color?

Fig. 10 title and description are not clear. Figure should be clearly comprehensible without reading the main text.  (Deregulated transcription factors in Candida tropicalis treatment – what treatment?)

Fig. 11 – there is no grey line for ergosterol in the diagram?

Table 1 is surplus – there is no need for table with two numbers inside, especially when this data is already given in the text.

I believe that this paper has a certain value purely because it provides a set of data on the expression levels of individual genes under the observed conditions. However, the figures and their descriptions / legends / titles should be edited and the sensationalism of the statements should be reduced. Data should be presented clearly and realistic, without exaggeration and much speculation.

Comments on the Quality of English Language

The grammar and the language used in the manuscript are of satisfactory quality, but the syntax of the sentences in the figure descriptions is often incomprehensible. It is not clear whether this is a consequence of insufficient knowledge of the language or imprecise scientific communication.

Author Response

Consulte el archivo adjunto

Reviewer 2 Report

Comments and Suggestions for Authors

Introduction:

  1. In line 38, the factors contributing to the surge in Candida species infections lack specific references or supporting evidence. Citations for each factor should be provided to strengthen the scientific basis of these claims.

  2. The transition to the Candida tropicalis section needs improvement. A clear and smooth transition from the general discussion of invasive candidiasis (IC) to the specific focus on Candida tropicalis would enhance the flow of the narrative.

  3. The term "remarkable relevance" in describing Candida tropicalis lacks specificity. Consider providing further clarification or justification for this term to enhance understanding.

  4. In line 100, the Gas Chromatography Conditions lack information on specific mass spectrometry parameters, such as ionization mode and specific ions monitored. Including these details would improve the reproducibility and clarity of the analysis.

  5. In line 205, the duration of exposure to ISO at the MIC for total ergosterol content determination (3 hours) lacks justification. Briefly explaining the choice of this duration would improve experimental design transparency.

Discussion:

  1. The discussion effectively highlights the significance of Candida tropicalis, drug resistance challenges, and the potential of Isopulegol (ISO) as an alternative antifungal agent. However, consider providing more context or importance of ergosterol in the context of antifungal targeting.

  2. Elaborate on the downregulation of methylation-related genes and its significance in the context of ISO's antifungal action. A more detailed exploration would enhance the interpretation of the findings.

  3. Emphasize the novelty of the study, particularly in identifying ERG6 as a potential target and the implications of ISO's action on transcription factors. Highlight the significance of these findings in the context of developing new antifungal agents.

  4. The manuscript could further emphasize practical applications and translational implications. Discuss how the identified molecular targets could lead to therapeutic interventions to enhance the article's impact.

Conclusion:

  1. Highlight the significance of adopting multi-faceted approaches in antifungal drug discovery. Discuss how targeting multiple aspects of fungal biology, as demonstrated by ISO, could be a promising strategy to counteract drug resistance and improve treatment outcomes.

Reviewer 3 Report

Comments and Suggestions for Authors

In the manuscript titled ''Transcriptional Reprogramming of Candida tropicalis in Response to Isoespintanol Treatment,'' authors presented a comprehensive analysis, as the continuation of their previous research, of the potential therapeutic targets in the battle against candidiasis and the mode of action of the monoterpene isoespintanol. In my sincere opinion, the presented manuscript represents significant insight into the battle with candidiasis and I really enjoyed reading the manuscript.
